# A Culture-Independent Analysis of the Microbiota of Female Interstitial Cystitis/Bladder Pain Syndrome Participants in the MAPP Research Network

**DOI:** 10.3390/jcm8030415

**Published:** 2019-03-26

**Authors:** J. Curtis Nickel, Alisa J. Stephens-Shields, J. Richard Landis, Chris Mullins, Adrie van Bokhoven, M. Scott Lucia, Jeffrey P. Henderson, Bhaswati Sen, Jaroslaw E. Krol, Garth D. Ehrlich

**Affiliations:** 1Department of Urology, Queen’s University, Kingston, ON K0H 2T0, Canada; jcn@queensu.ca; 2Department of Biostatistics and Epidemiology, Perelman School of Medicine, University of Pennsylvania, Philadelphia, PA 19104, USA; alisaste@pennmedicine.upenn.edu (A.J.S.-S.); jrlandis@pennmedicine.upenn.edu (J.R.L.); 3National Institute of Diabetes and Digestive and Kidney Diseases, National Institutes of Health, Bethesda, MD 20892, USA; mullinsc@extra.niddk.nih.gov; 4Department of Pathology, University of Colorado Anschutz Medical Campus, Aurora, CO 80045, USA; adrie.vanbokhoven@ucdenver.edu (A.v.B.); scott.lucia@ucdenver.edu (M.S.L.); 5Department of Medicine, Division of Infectious Diseases, Washington University School of Medicine, St. Louis, MO 63110, USA; hendersonj@wustl.edu; 6Departments of Microbiology & Immunology; Drexel University College of Medicine, Philadephia, PA 19102, USA; bs563@drexel.edu (B.S.); jek322@drexel.edu (J.E.K.); 7Department of Otolaryngology-Head and Neck Surgery, Drexel University College of Medicine, Philadelphia, PA 19102, USA; 8c/o Department of Biostatistics and Epidemiology, Perelman School of Medicine, University of Pennsylvania, Philadelphia, PA 19104, USA; mapp-bio@lists.upenn.edu

**Keywords:** microbiota, microbiome, infection, interstitial cystitis, bladder pain syndrome

## Abstract

We surveyed urine microbiota of females diagnosed with interstitial cystitis/bladder pain syndrome (IC/BPS) and matched control participants enrolled in the National Institutes of Health (NIH) Multidisciplinary Approach to the Study of Chronic Pelvic Pain (MAPP) Research Network using the culture-independent methodology. Midstream urine specimens were analyzed with the Plex-ID molecular diagnostic platform that utilizes polymerase chain reaction–electrospray ionization–time-of-flight–mass spectrometry (PCR-ESI-TOF MS) to provide a comprehensive identification of bacterial and select fungal species. IC/BPS and control participants were evaluated for differences (presence, diversity, and abundance) in species and genus. Urine specimens obtained from 181 female IC/BPS and 182 female control participants detected a total of 92 species (41 genera). Mean (SD) species count was 2.49 (1.48) and 2.30 (1.28) among IC/BPS and control participants, respectively. Overall species composition did not significantly differ between IC/BPS and control participants at any level (*p* = 0.726 species level, *p* = 0.222 genus level). IC/BPS participants urine trended to an overabundance of *Lactobacillus gasseri* (*p* = 0.09) detected but had a lower prevalence of *Corynebacterium* compared with control participants (*p* = 0.002). The relative abundance data analysis mirrored the prevalence data differences with no significant differences in most species or genus abundance other than *Lactobacillus gasseri* and *Corynebacterium* (*p* = 0.08 and *p* = 0.001, respectively). No cause and/or effect conclusion can be drawn from this observation, but it suggests that a more comprehensive evaluation (vaginal, bowel, catheterized bladder and/or tissue-based specimens) of the lower urinary tract microbiota in IC/BPS patients is warranted.

## 1. Introduction

Interstitial cystitis/bladder pain syndrome (IC/BPS) is an enigmatic urological condition associated with bladder pain and urinary frequency and urgency in the absence of typical urinary tract infection [1]. Based on the most conservative USA estimates [2], approximately 3.8 million women and 1.4 million men suffer from this condition, which severely impacts on the quality of life [3]. Management is difficult, and a cure remains elusive because the etiology and pathophysiology are unknown [1,4]. The diagnosis of IC/BPS relies on a clinical evaluation that confirms the symptoms that characterize IC/BPS (bladder pain with urinary storage symptoms) and rules out other confusable diseases, such as infection, neurologic causes, malignancy, urinary stones, etc. [1]. Contemporary treatment of the typical patient with IC/BPS includes initial conservative management (education, diet changes, stress management, relaxation/stretching exercise, and pelvic floor physiotherapy) with the usual addition of further medical (and surgical) treatments to address the pain and urinary symptoms. The clinical trial evidence shows that one specific therapy does not work for all, but acceptable symptom control can be achieved in many patients by using single or multimodal therapy individualized for each patient [1,4]. Oral medications used for IC/BPS include amitriptyline, cimetidine, hydroxyzine, pentosanpolysulfate sodium while intravesical instillations of dimethylsulfoxide (DMSO), heparin or lidocaine are employed when medical therapy is not successful. Some patients with ongoing symptoms respond to minor bladder surgery (cauterization of inflammatory lesions called Hunner lesions or injection of botulinum toxin A) while a few treatment refractory patients are considered for a neuromodulation (neurostimulation) or immune modulation (cyclosporine A). Unfortunately, many patients either do not respond or respond poorly to these approaches and differentiation of subsets of IC/BPS patients based on etiology or mechanisms are required to improve our therapeutic outcomes.

Although active urinary tract infection excludes the diagnosis, and empiric antibiotic therapy is typically unhelpful, a bacterial etiology has never been excluded as a mechanism in IC/BPS, and such an association has been suggested in a small but perhaps significant number of IC/BPS patients [5,6,7,8,9,10]. The microbiologic diagnosis of infection in the bladder has traditionally been based on cultivation techniques in which bacteria are grown from voided urine spread on culture plates, which does not have the nutritive and environmental conditions required to support the growth of many microorganisms. We now understand that these traditional tools used to study bacteria, not only in IC/BPS patients but also in patients with presumed bacterial cystitis, are inadequate as a means to survey the microorganisms present in patient samples. Molecular [11], as well as novel [12] and enhanced culture [13] techniques, have been developed that improve the detection, identification, and quantification of microorganisms in urine. Such techniques have clearly demonstrated that the lower urinary tract is not sterile [13]. The bladder microbiome in healthy females [14,15] is very variable between individuals, with many showing a lack of diversity with the domination of one bacterial species or genus while others have more diverse microbiomes with no specific genus dominating. The most common genera are *Lactobacillus*, *Gardnerella*, *Streptococcus,* and/or *Staphylococcus*, but *Enterobacteriaceae*, typically considered uropathogenic, can also be present in healthy bladders of asymptomatic women.

Studies have shown that differences in the microbiome of the bladder may be associated with lower urinary tract symptoms, such as urgency urinary incontinence [16,17,18] but others have produced conflicting microbiological data in IC/BPS patients [19]. We have used Ibis T-5000 Universal Biosensor technology [20] which employs a polymerase chain reaction–electron spray ionization–time-of-flight–mass spectrometry (PCR-ESI-TOF MS) coupled to a dynamic relational database, to provide comprehensive identification of all bacterial species present at >1–3% of the microbiome to examine the microbiota of men with chronic prostatitis/chronic pelvic pain syndrome (CP/CPPS) [21] and flare episodes in women with IC/BPS [22]. We hypothesized that similar PCR-ESI-TOF MS technology might identify differences in the microbiota of the lower urinary tract between female IC/BPS and control participants (without IC/BPS) enrolled in the National Institutes of Health (NIH) Multidisciplinary Approach to the Study of Chronic Pelvic Pain (MAPP) Research Network [23].

## 2. Experimental Section

### 2.1. Participants and Specimens

The MAPP Research Network’s initial central clinical study, the Trans-MAPP epidemiology/phenotyping (EP) Study, recruited participants with urologic chronic pelvic pain syndrome (UCPPS), an umbrella term used to include both IC/BPS (primarily females) and CP/CPPS (males), for comprehensive baseline phenotyping and longitudinal follow-up of the treated history of related symptoms, with standardized data acquisition and analysis and biological sample collection across network sites. All network sites received individual institution IRB/REB approval. Clinical parameters included the interstitial cystitis symptom index (ICSI), the genitourinary pain index (GUPI) and documentation of other associated chronic pain conditions. In addition, the study enrolled controls (individuals with no IC/BPS), for the same baseline phenotyping assessments [23].

Inclusion criteria for female IC/BPS study participants for the Trans-MAPP EP Study included: (1) a diagnosis of IC/BPS, with urologic symptoms present a majority of the time during the most recent 3 months; (2) at least 18 years old; (3) reporting a non-zero score for bladder and/or pelvic region pain, pressure or discomfort during the past 2 weeks; and (4) appropriate consent. Exclusion criteria have been described [24]. Controls were recruited to be age- and sex-matched to IC/BPS patients. Inclusion and Exclusion criteria are described in Appendix B. Further details of the study design, including descriptions of the study population enrollment criteria and disease-specific questionnaires, are available [23,24].

The current study details the analysis of clinical data, and associated midstream urine specimens collected at the in-clinic baseline phenotyping visit from female IC/BPS and control participants enrolled in the Trans-MAPP EP Study. IC/BPS patients with a positive urine culture (i.e., traditional uropathogen(s) detected in midstream collected urine employing traditional culture technique) were excluded from the analysis as per Trans-MAPP EP protocol. Previous antimicrobial therapy was not an exclusion criterion.

### 2.2. Specimen Handling

Following in-clinic collection of urine specimens using standardized collection kits at the MAPP Network Discovery sites, specimens were transferred to 50 mL conical tubes and frozen at −80 °C (85% of specimens were frozen within 15 min and more than 95% within 30 min) then shipped to the central MAPP network tissue analysis and technology core (TATC). The specimens were then thawed, thoroughly mixed, and aliquoted into 1 and 3 mL aliquots and refrozen at −80 °C until use. Three milliliter frozen aliquots were transferred to the Center for Genomic Sciences at Drexel University College of Medicine in Philadelphia, PA for microbial analyses.

### 2.3. DNA Extraction and Ibis Eubacterial and Fungal Domain Assays on the PLEX-ID

The Plex-ID molecular diagnostic platform and infectious agent diagnostic kits (Abbott Molecular, Des Plaines, IL, USA) are technically identical to the Ibis T-5000 Universal Biosensor Analysis platform and kits [20] (Ibis Biosciences, Carlsbad CA USA) that we previously used for the MAPP Network microbiome studies [21,22]; the only difference is the manufacturer of the instrument has changed. The advantages, limitations, and comparisons to other culture-enhanced and molecular diagnostic techniques of these methods have been previously described [20,21,22]. In brief, total DNA was extracted from all urine samples, and microbial (i.e., bacterial and fungal) DNAs were amplified by polymerase chain reaction (PCR) using the 16 primer pair BAC (Bacteria, Antibiotic resistance genes and Candida) detection systems developed by Ibis as described [14,15,16]. The individual amplicons were “weighed” using the PLEX-ID instrumentation by electrospray ionization—time-of-flight—mass spectrometry (ESI-TOF-MS) which reports out the molecular mass. The amplicon masses were then used to determine their exact base compositions as a particular mass can only be produced by a single combination of the four nucleotides (A, C, G, T). The taxonomic identities of the amplicons were then revealed using a database containing base composition data on virtually all bacterial/fungal species sequenced to date. Comparison of this technique with other non-culture methods and details of the methodology are beyond the scope of this paper (details are available in [20,21,22] and Appendix C).

### 2.4. Statistical Analysis

Demographic characteristics and relevant clinical features were compared between IC/BPS and control participants by chi-square tests. Chao1 [25] and Shannon’s index [26] were compared between female participants and controls by the Wilcoxon’s rank-sum test. Differences in the representation of individual taxa were tested using logistic regression for presence or absence and Wilcoxon’s rank-sum test for relative abundance. Differences in the overall microbial composition for IC/BPS versus controls were assessed by permutational multivariate analysis (PERMANOVA) [27]. This procedure is a nonparametric analogue of multivariate ANOVA that uses resampling for inference. The abundance of particular taxa for each subject is converted into a numerical matrix from which distance matrices are calculated and compared between groups according to a selected distance measure. The Bray–Curtis and Jaccard distances were chosen as the basis of this analysis, using samples that contained at least one detected species. Testing was conducted at the species and genus levels. In the univariate analyses comparing one species at a time between participant groups, we restricted comparisons to species (or genera) present in 10 or more participants to reduce the likelihood of underpowered analyses. Tests of individual taxa were adjusted for multiple comparisons by controlling the false discovery rate (FDR) [28].

## 3. Results

### 3.1. Demographic Data

Urine specimens and extensive clinical data were obtained from 181 female IC/BPS and 182 female control participants at the baseline visit of the Trans-MAPP EP Study. Demographic data are detailed in Table 1.

### 3.2. Clinical Data

All IC/BPS patients met a clinically-defined IC/BPS diagnostic criteria (i.e., unpleasant sensation of pain, pressure, or discomfort, perceived to be related to the bladder and/or pelvic region, associated with lower urinary tract symptoms in the most recent 3 months before enrollment). In the IC/BPS participant group, the mean (SD) ICSI score was 10.9 (4.4) while the female GUPI score was 26.8 (4.7). Baseline clinical data are summarized in Table 2.

### 3.3. Species Data

A total of 92 species (41 genera) were detected in the urine specimens. There was substantial overlap in the species composition between IC/BPS and Control participants (Figure 1, Table 3). Overall, there were 332 participants (91.4%) in which at least 1 species was detected. In IC/BPS participants it was 161 (89.2%); in control participants it was 171 (94.0%). Mean (SD) species count per person was 2.5 (1.5) and 2.3 (1.3) among IC/BPS and control participants, respectively. Shannon’s index and Chao1 did not differ by cohort at the species level (*p* = 0.18 Chao1; *p* = 0.15 Shannon’s index; Table 4). Overall species composition did not significantly differ between IC/BPS and control participants at any level by either Bray–Curtis or Jaccard distances (*p* = 0.73 Bray–Curtis, *p* = 0.62 Jaccard species level; Figure 2, Table 5). *Lactobacillus gasseri* trended as more prevalent and over-abundant in the IC/BPS participants compared to controls, but did not reach statistical significance at (*p* = 0.09 and 0.08, respectively).

### 3.4. Genus Data

Analysis of the genus-level data is shown in Table 4 and Table 6 and Figure 3. Testing of individual genera showed the only significant difference was with *Corynebacterium* sp., showing greater relative abundance among controls compared to IC/BPS participants (*p* = 0.001), a finding which was consistent with the presence/absence prevalence data (*p* = 0.002).

## 4. Discussion

*Lactobacillus gasseri* was slightly more prevalent in IC/BPS, while there was lower prevalence of *Corynebacterium* among IC/BPS participants compared to controls. Moreover, the only differences in relative abundances were a lower abundance of *Corynebacterium* (genus level) and marginally higher abundance of *Lactobacillus gasseri* (species level) in the IC/BPS participant population compared with the control population. We did not identify any other significant differences in species diversity or overall species composition between IC/BPS and control participants at either the genus or the species level. Although we did not note any statistical difference in species composition, we did identify 29 species unique to the IC/BPS group that were not identified in the control group. These species included bacteria that are considered potentially uropathogenic (such as *Escherichia coli*, *Proteus mirabilis*, *Pseudomonas aeruginosa,* etc.) as well as potentially virulent organisms (such as *Francisella tularensis*, *Mycoplasma hyorhinis*, *Helicobacter hepaticus*, *Clostridium perfringens*, *Candida dubliniensis,* etc.). While our study cannot and does not implicate these organisms in the etiology of IC/BPS, we cannot discount their role as potential etiologic organisms in some individuals, either by themselves or in some deleterious combination (dysbiosis).

The lower urinary tract and bladder are not sterile, as has been traditionally assumed, but contain a resident microbiota, including organism’s refractory to standard culturing techniques [12,14,15]. There have been some attempts to employ earlier generation molecular diagnostic techniques to search for the presence of causative organisms in patients with negative urine cultures and a diagnosis of IC/BPS [5,6,7,8,9,10]. However, findings have been inconclusive [19]. The most recent study which examined midstream urine collected from eight patients with a diagnosis of culture negative IC with 16S ribosomal DNA (rDNA) PCR and 454-pyrosequencing identified 31 different genera [29]. Compared to historical controls (data from previous studies), the authors identified a significantly higher proportion of IC patients with *Lactobacillus* and a decrease in bacterial diversity. *Lactobacilli gasseri* has been detected more frequently in a urinary incontinence cohort (UUI) cohort compared to controls [16]. These studies, along with our findings, suggest a possible role for *Lactobacillus* (specifically *Lactobacilli gasseri*) in some patients with lower urinary tract symptoms. The finding of a decreased prevalence of *Corynebacterium* in the IC/BPS group may also be implicated. While it is difficult to explain any relevant mechanism, one could hypothesize that *Corynebacterium* might be beneficial or even protective in the polymicrobial bladder microbiome of asymptomatic females. While not directly related to the urinary tract microbiome, a recent report observed microbiome changes in the stool of IC/BPS patients compared to asymptomatic controls [30].

As with all the urologic microbiome studies published to date, limitations of our study should be noted. The technique we used can detect micro-organisms but not microbial memory or history (persisting pain after eradication of bacteria) and does not inform on virulence. The PCR-ESI-TOF-MS technology identifies up to 97% of the total microbiome, meaning that we could miss some rare or very low number micro-organisms. Our technique only identifies the fungus *Candida* sp. and *Saccharomyces* sp. and no viruses. The specimens did not undergo other DNA sequencing methodology or enhanced culture protocols, which conceivably may have demonstrated other results. Perhaps the greatest limitation of our study was the sample type; only a biopsy of the bladder could rule out a microbial biofilm attached to or within the bladder wall. We examined voided urine which would include mixed populations of microbes from kidney, bladder, urethral, and vaginal origins. While catheterized urine would allow for a more direct analysis of microbes within the bladder lumen, this technique examined only broad differences in the lower urinary tract microbiome between IC/BPS and control participants, not simply populations specific to the bladder. Such a study would require at least bladder catheterization and preferably bladder biopsy, procedures that were not available to us in this large trans-MAPP cohort. This cross-sectional analysis of the baseline microbiomes between IC/BPS participants and controls does not rule out the possibility that various organisms identified in IC/BPS might be implicated in the development and/or maintenance of various phenotypic subgroups. Finally, we were not able to factor in the implication of not adjusting for previous antibiotic therapy on the bladder microbiota. One of the major strengths of our study includes the fact that this is the largest, most comprehensively studied cohort of IC/BPS patients compared to matched control subjects. The current ongoing MAPP-2 study includes broad, diverse, and longitudinal biological samples collections (including urine, fecal and vaginal microbiome), as well as longitudinal, information (including antibiotic use) becoming available in respect to associated clinical data and other phenotypic data for future sub-grouping analysis.

## 5. Conclusions

Assessment of culture independent microbiological data from participants enrolled in the MAPP Network’s central clinical phenotyping study, the Trans-MAPP EP Study, has identified possible associations of the urologic microbiome and IC/BPS. As well as identification of potentially unique species to IC/BPS, potential differences noted in *Lactobacillus gasseri* and *Corynebacterium* between IC/BPS and control participants complements previous studies for IC and UUI and suggests possible microbiome influence on lower urinary tract clinical phenotypes. The observation may suggest a deleterious impact of *Lactobacillus gasseri* and perhaps a beneficial or protective influence of *Corynebacterium*. However, this study cannot be used to infer cause and/or effect. Longitudinal studies examining relationships between the urologic microbiome and IC/BPS symptom patterns and clinical sub-phenotypes are ongoing as part of the ongoing MAPP Network research program.

## Figures and Tables

**Figure 1 jcm-08-00415-f001:**
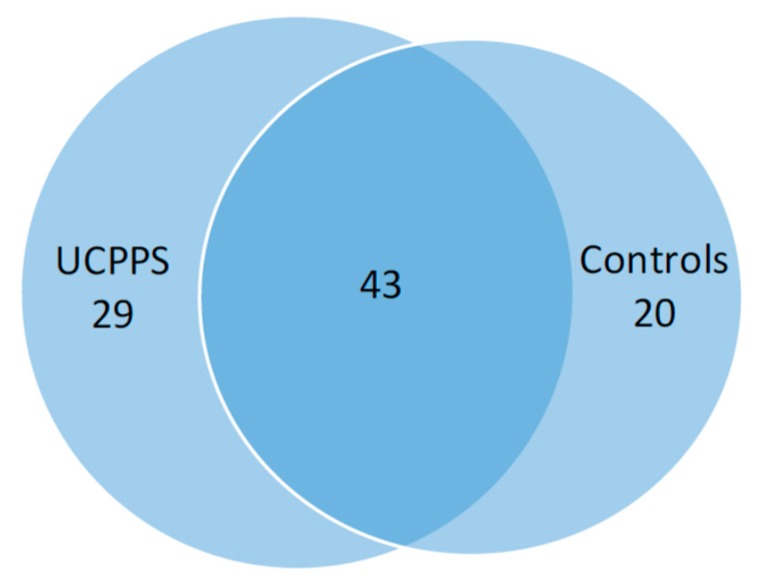
Venn diagram of overlapping species. A total of 92 species were detected with 29 species in IC/BPS only, 20 species in controls only, and 43 in both IC/BPS and controls

**Figure 2 jcm-08-00415-f002:**
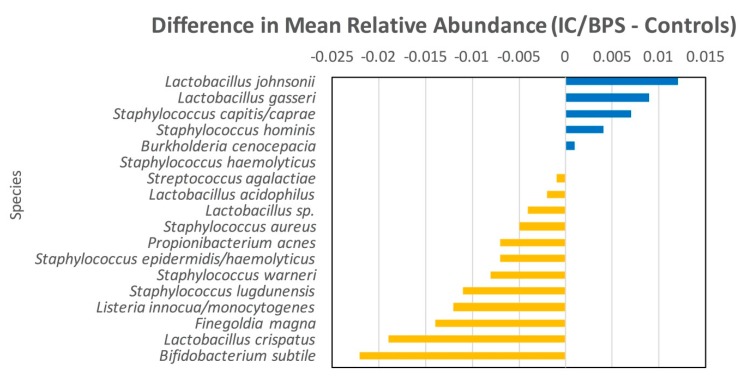
Differences in mean relative abundance at the species level (IC/BPS minus controls). Blue indicates more abundant among urologic chronic pelvic pain syndrome (UCPPS). Yellow is more abundant among controls.

**Figure 3 jcm-08-00415-f003:**
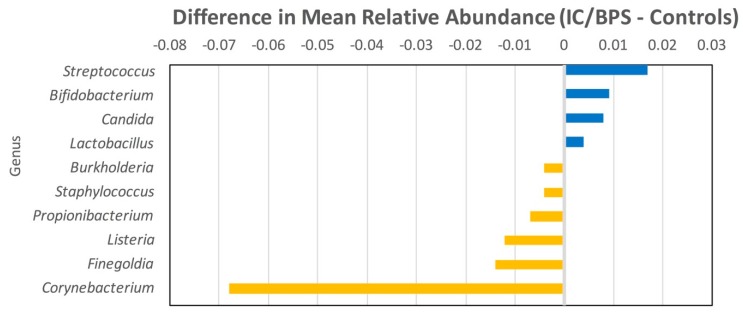
Differences in mean relative abundance at the genus level (IC/BPS minus controls) Blue indicates more abundant among UCPPS. Yellow is more abundant among controls.

**Table 1 jcm-08-00415-t001:** Baseline Demographic Data.

Parameter	Category	*IC/BPS	Controls	Total	*p*
Number of Participants	*n* (%)	181	182	363	
Clinical Site	Northwestern U	17 (9.4%)	22 (12.1%)	39 (10.7%)	0.906
	UCLA	25 (13.8%)	24 (13.2%)	49 (13.5%)	
	U of Iowa	36 (19.9%)	29 (15.9%)	65 (17.9%)	
	U of Michigan	33 (18.2%)	33 (18.1%)	66 (18.2%)	
	U of Washington	23 (12.7%)	29 (15.9%)	52 (14.3%)	
	Wash U St. Louis	39 (21.5%)	37 (20.3%)	76 (20.9%)	
	Stanford U	8 (4.4%)	8 (4.4%)	16 (4.4%)	
Age Group	<35 years	76 (42.0%)	81 (44.5%)	157 (43.3%)	0.869
	35–50 years	50 (27.6%)	50 (27.5%)	100 (27.5%)	
	50+ years	55 (30.4%)	51 (28.0%)	106 (29.2%)	
Race	White	165 (91.2%)	137 (75.3%)	302 (83.2%)	<0.001
	Black	5 (2.8%)	25 (13.7%)	30 (8.3%)	
	Asian	2 (1.1%)	10 (5.5%)	12 (3.3%)	
	Multi Race	3 (1.7%)	5 (2.7%)	8 (2.2%)	
	Other	5 (2.8%)	4 (2.2%)	9 (2.5%)	
	Unknown	1 (0.6%)	1 (0.5%)	2 (0.6%)	
Ethnicity	Hispanic	11 (6.1%)	11 (6.0%)	22 (6.1%)	1.000
	Non-Hispanic	170 (93.9%)	171 (94.0%)	341 (93.9%)	

*IC/BPS: Interstitial Cystitis/Bladder Pain Syndrome.

**Table 2 jcm-08-00415-t002:** Baseline clinical data.

Parameter	Category	IC/BPS	Controls	Total	*p*
Number of Participants	*n* (%)	181	182	363	
Self-reported IC diagnosis	No	24 (13.3%)	169 (92.9%)	193 (53.2%)	<0.001
	Yes	157 (86.7%)	2 (1.1%)	159 (43.8%)	
	Missing		11 (6.0%)	11 (3.0%)	
Meet MAPP IC/BPS Criteria	Yes	181 (100.0%)		181 (49.9%)	
	Missing		182 (100.0%)	182 (50.1%)	
IC diagnosis from Rice form	No	64 (35.4%)	172 (94.5%)	236 (65.0%)	<0.001
	Yes	117 (64.6%)	10 (5.5%)	127 (35.0%)	
Associated Chronic Pain Syndrome	None	105 (58.0%)	110 (60.4%)	215 (59.2%)	0.670
	Any Syndrome	76 (42.0%)	72 (39.6%)	148 (40.8%)	
Interstitial Cystitis Symptom Index (ICSI)		10.9 (4.4)	3.0 (3.2)	7.0 (5.5)	<0.001
Genitourinary Pain Index (GUPI)		26.6 (8.7)	4.7 (7.4)	15.7 (13.6)	<0.001
Meds for urologic or pelvic pain symptoms	No	37 (20.4%)	176 (96.7%)	213 (58.7%)	<0.001
	Yes	144 (79.6%)	6 (3.3%)	150 (41.3%)	
Pain medication class	None	36 (19.9%)	121 (66.5%)	157 (43.3%)	<0.001
	Peripheral	42 (23.2%)	24 (13.2%)	66 (18.2%)	
	Central	81 (44.8%)	28 (15.4%)	109 (30.0%)	
	Opioid	22 (12.2%)	9 (4.9%)	31 (8.5%)	

**Table 3 jcm-08-00415-t003:** Unique and overlapping species identified in IC/BPS and control subjects.

Species unique to IC/BPS	Species unique to Controls	Species found in both:
*1. Acinetobacter grimontii*	*1. Bordetella parapertussis*	*1. Bacteroides ureolyticus*
*2. Akkermansia muciniphila*	*2. Burkholderia sp.*	*2. Bifidobacterium inopinatum*
*3. Bacillus sp.*	*3. Clostridium sp.*	*3. Bifidobacterium longum*
*4. Bifidobacterium bifidum*	*4. Enterococcus faecium*	*4. Bifidobacterium subtile*
*5. Bifidobacterium infantis*	*5. Haemophilus influenzae*	*5. Bordetella bronchiseptica*
*6. Borrelia turicatae*	*6. Klebsiella oxytoca*	*6. Burkholderia cenocepacia*
*7. Candida dubliniensis*	*7. Klebsiella pneumoniae*	*7. Candida albicans*
*8. Clostridium perfringens*	*8. Lactobacillus collinoides*	*8. Candida glabrata*
*9. Escherichia coli*	*9. Ochrobactrum anthropi*	*9. Corynebacterium diphtheriae*
*10. Helicobacter hepaticus*	*10. Pasteurella multocida*	*10. Corynebacterium jeikeium*
*11. Lactobacillus casei*	*11. Pediococcus pentosaceus*	*11. Corynebacterium pseudodiphtheriticum*
*12. Lactobacillus helveticus*	*12. Pseudomonas stutzeri*	
*13. Lactobacillus reuteri*	*13. Serratia marcescens*	
*14. Lactococcus lactis*	*14. Streptococcus mutans*	
*15. Proteus mirabilis*	*15. Streptococcus sanguinis*	
*16. Pseudomonas aeruginosa*	*16. Treponema denticola*	
*17. Salmonella enterica*	*17. Ureaplasma urealyticum*	
*18. Staphylococcus intermedius*	*18. Francisella philomiragia*	
*19. Streptococcus dysgalactiae*	*19. Corynebacterium striatum*	
*20. Streptococcus porcinus*	*20. Microbacterium sp.*	
*21. Streptococcus pyogenes*		
*22. Bacteroides vulgatus*		
*23. Bifidobacterium pseudocatenulatum*		
*24. Francisella tularensis*		
*25. Mycoplasma hyorhinis*		
*26. Paracoccus denitrificans*		
*27. Staphylococcus sp.*		
*28. Micrococcus lylae*		
*29. Tetragenococcus halophilus*		

**Table 4 jcm-08-00415-t004:** Chao1 and Shannon’s index at the species and genus level.

Taxonomic Level	Index	Controls		UCPPS		*p* ^1^
		*n*	Mean (SD)	*n*	Mean (SD)	
Species	Chao1	182	2.3 (1.3)	181	2.5 (1.5)	0.18
	Shannon	182	0.2 (0.3)	181	0.3 (0.3)	0.15
Genus	Chao1	182	1.9 (1.1)	181	2 (1.1)	0.41
	Shannon	182	0.2 (0.3)	181	0.2 (0.3)	0.38

^1^ Calculated by Wilcoxon’s rank-sum test.

**Table 5 jcm-08-00415-t005:** Species-level differences in prevalence and relative abundance for urologic chronic pelvic pain syndrome (UCPPS) participants compared to Controls.

	Controls		UCPPS		Associate: Prevalence		Association: Relative Abundance	
taxa	Prevalence	Mean (SD) Relative Abundance	Prevalence	Mean (SD) Relative Abundance	OR (95% CI)	*p* ^1^	Mean Difference	*p* ^2^
*Staphylococcus hominis*	20/182 (11%)	0.01 (0.055)	29/181 (16%)	0.014 (0.091)	1.55 (0.84,2.85)	0.163	0.004	0.216
*Staphylococcus lugdunensis*	6/182 (3.3%)	0.012 (0.106)	6/181 (3.3%)	0.002 (0.016)	1.01 (0.32,3.18)	0.992	−0.011	0.996
*Staphylococcus warneri*	10/182 (5.5%)	0.011 (0.091)	4/181 (2.2%)	0.003 (0.043)	0.39 (0.12,1.26)	0.116	−0.008	0.098
*Streptococcus agalactiae*	11/182 (6%)	0.009 (0.068)	11/181 (6.1%)	0.008 (0.078)	1.01 (0.42,2.38)	0.989	−0.001	0.972
*Bifidobacterium subtile*	26/182 (14.3%)	0.128 (0.322)	26/181 (14.4%)	0.106 (0.282)	1.01 (0.56,1.81)	0.982	−0.022	0.872
*Burkholderia cenocepacia*	12/182 (6.6%)	0.018 (0.128)	8/181 (4.4%)	0.019 (0.128)	0.66 (0.26,1.64)	0.367	0.001	0.389
*Finegoldia magna*	10/182 (5.5%)	0.018 (0.113)	12/181 (6.6%)	0.004 (0.029)	1.22 (0.51,2.9)	0.651	−0.014	0.703
*Lactobacillus acidophilus*	21/182 (11.5%)	0.062 (0.223)	20/181 (11%)	0.06 (0.22)	0.95 (0.5,1.82)	0.883	−0.002	0.878
*Lactobacillus crispatus*	54/182 (29.7%)	0.277 (0.434)	51/181 (28.2%)	0.258 (0.419)	0.93 (0.59,1.46)	0.753	−0.019	0.415
*Lactobacillus gasseri*	5/182 (2.7%)	0.015 (0.11)	12/181 (6.6%)	0.024 (0.122)	2.51 (0.87,7.29)	0.090	0.009	0.084
*Lactobacillus johnsonii*	45/182 (24.7%)	0.027 (0.127)	51/181 (28.2%)	0.038 (0.146)	1.19 (0.75,1.91)	0.456	0.012	0.329
*Lactobacillus sp.*	11/182 (6%)	0.059 (0.233)	11/181 (6.1%)	0.055 (0.218)	1.01 (0.42,2.38)	0.989	−0.004	0.96
*Listeria innocua/monocytogenes*	8/182 (4.4%)	0.013 (0.098)	5/181 (2.8%)	0.001 (0.016)	0.62 (0.2,1.93)	0.407	−0.012	0.39
*Propionibacterium acnes*	10/182 (5.5%)	0.036 (0.173)	12/181 (6.6%)	0.03 (0.145)	1.22 (0.51,2.9)	0.651	−0.007	0.686
*Staphylococcus aureus*	6/182 (3.3%)	0.006 (0.051)	4/181 (2.2%)	0.001 (0.008)	0.66 (0.18,2.39)	0.560	−0.005	0.524
*Staphylococcus capitis/caprae*	8/182 (4.4%)	0.009 (0.077)	7/181 (3.9%)	0.016 (0.108)	0.88 (0.31,2.47)	0.801	0.007	0.814
*Staphylococcus epidermidis/haemolyticus*	57/182 (31.3%)	0.05 (0.184)	61/181 (33.7%)	0.042 (0.171)	1.11 (0.72,1.73)	0.628	−0.007	0.816
*Staphylococcus haemolyticus*	10/182 (5.5%)	0.004 (0.037)	11/181 (6.1%)	0.004 (0.031)	1.11 (0.46,2.69)	0.812	0.000	0.819

^1^ Determined by logistic regression of species presence onto cohort. ^2^ Determined by Wilcoxon’s rank-sum test.

**Table 6 jcm-08-00415-t006:** Genus Level Differences in Prevalence and Relative Abundance for UCPPS participants compared to Controls.

	Controls		UCPPS		Associate: Prevalence		Association: Relative Abundance	
Taxa	Prevalence	Mean (SD) Relative Abundance	Prevalence	Mean (SD) Relative Abundance	OR (95% CI)	*p* ^1^	Mean Difference	*p* ^2^
*Bifidobacterium*	28/182 (15.4%)	0.138 (0.333)	35/181 (19.3%)	0.148 (0.327)	1.32 (0.76,2.28)	0.321	0.009	0.461
*Burkholderia*	13/182 (7.1%)	0.023 (0.147)	8/181 (4.4%)	0.019 (0.128)	0.6 (0.24,1.49)	0.271	−0.004	0.283
*Candida*	5/182 (2.7%)	0 (0.004)	11/181 (6.1%)	0.009 (0.078)	2.29 (0.78,6.73)	0.132	0.008	0.119
*Corynebacterium*	23/182 (12.6%)	0.08 (0.248)	6/181 (3.3%)	0.012 (0.085)	0.24 (0.09,0.6)	0.002*	−0.068	0.001 *
*Finegoldia*	10/182 (5.5%)	0.018 (0.113)	12/181 (6.6%)	0.004 (0.029)	1.22 (0.51,2.9)	0.651	−0.014	0.703
*Lactobacillus*	104/182 (57.1%)	0.457 (0.473)	109/181 (60.2%)	0.461 (0.461)	1.14 (0.75,1.72)	0.552	0.004	0.822
*Listeria*	8/182 (4.4%)	0.013 (0.098)	5/181 (2.8%)	0.001 (0.016)	0.62 (0.2,1.93)	0.407	−0.012	0.39
*Propionibacterium*	10/182 (5.5%)	0.036 (0.173)	12/181 (6.6%)	0.03 (0.145)	1.22 (0.51,2.9)	0.651	−0.007	0.686
*Staphylococcus*	101/182 (55.5%)	0.107 (0.258)	99/181 (54.7%)	0.103 (0.263)	0.97 (0.64,1.46)	0.879	−0.004	0.511
*Streptococcus*	17/182 (9.3%)	0.018 (0.099)	26/181 (14.4%)	0.035 (0.156)	1.63 (0.85,3.12)	0.141	0.017	0.141

^1^ Determined by logistic regression of species presence onto cohort. ^2^ Determined by Wilcoxon’s rank-sum test (see text). * Statistically significant.

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
