# Peer review of "A Culture-Independent Analysis of the Microbiota of Female Interstitial Cystitis/Bladder Pain Syndrome Participants in the MAPP Research Network"

_jcm, 2019, doi:10.3390/jcm8030415_

Round 1
Reviewer 1 Report
In the following manuscript authors have surveyed urine microbiota of females diagnosed with IC/BPS and matched control participants enrolled in the NIH MAPP Research Network using culture-independent methodology. They have highlighted that IC/BPS participants urine trended to an overabundance of Lactobacillus gasseri but had lower prevalences of Corynebacterium compared to control participants. However, the only significant difference between IC/BPS and matched control participants was reached concerning the abundance of Corynebacterium. I was impressed with the large number of patients who were enrolled in this study but there are main points in the manuscript that have to be clarified and better explained. Please see my comments below:
Line 43: Infection was repeated two times
Line 78: Authors have to include the exclusion criteria in the text (i.e. as supplementary table). Do they enrolled patients that have consumed antibiotics in the last 3 months? It can be considered as important limiting factor for the whole study. Further information are needed.
Line 139: Authors have to specific the abbreviations ICSI and GUPI used in the table 2. They were not reported within the main text.
Line 156: In the figure 2, authors have put in relevance the differences in mean relative abundance at the species levels between IC/BPS-Controls. Authors have described an higher abundance of Lactobacillus johnsonii as compared with the control group. However, several studies in rodents have shown the role of Lactobacillus johnsonii to prevent gastritis, reduce the level of pro-inflammatory cytokines, intestinal, extra-intestinal and systemic pro-inflammatory immune responses, diabetes. For this reason, I was wondering how the authors can interpret those results. Do they check the body weight, inflammatory markers, insulin and other parameters linked to metabolic and inflammatory disorders of those patients? Further information are needed.
In addition, authors described a lower abundance of Corynebacterium in the IC/BPS participant population compared to control population. In the literature, other authors have described that Corynebacterium species are common in polymicrobial infections. Further info and explanation are needed.
Line 158: To check the table 3, there is a mistake “9” has to be replaced with (.
Line 198: Authors have mentioned within the main text that a recent report observed microbiome changes in the stool of IC/BPS patients compared to asymptomatic controls. Why did not the authors decide to do the same analysis? Do they collected the stool material of those patients?
Author Response
February 20, 2019
Marta Varo, MSc
Genes Editorial Office, MDPI
Re: jcm-450936
Dear Ms. Varo,
The authors would like to thank the editors and the 4 reviewers for their careful and critical reviews of our manuscript jcm-450936 “A Culture-Independent Analysis of the Microbiota of Female Interstitial Cystitis/Bladder Pain Syndrome (IC/BPS) Participants in the MAPP Research Network”. We have answered each of the reviewer (and editors) queries below and have made the required changes in the revised manuscript (tracked in Word).
Reviewer 1
In the following manuscript authors have surveyed urine microbiota of females diagnosed with IC/BPS and matched control participants enrolled in the NIH MAPP Research Network using culture-independent methodology. They have highlighted that IC/BPS participants urine trended to an overabundance of Lactobacillus gasseri but had lower prevalences of Corynebacteriumcompared to control participants. However, the only significant difference between IC/BPS and matched control participants was reached concerning the abundance of Corynebacterium. I was impressed with the large number of patients who were enrolled in this study but there are main points in the manuscript that have to be clarified and better explained. Please see my comments below:
Line 43: Infection was repeated two times
Reply: The reviewer has pointed out that the first sentence ended with the word “infection” while the second sentence started with the same word. We have now expanded the Introduction section (as per reviewer’s comments) to include a more detailed description of IC/BPS and the bladder microbiome including references. The two sentences are now separated by the expanded IC/BPS explanation and the wording has been revised.
Line 78: Authors have to include the exclusion criteria in the text (i.e. as supplementary table). Do they enrolled patients that have consumed antibiotics in the last 3 months? It can be considered as important limiting factor for the whole study. Further information is needed.
Reply: We indicated that this information is available (and included the references) but agree with referee that it would be helpful to add this information as an appendix which we have done (Appendix B). We have renumbered the other supplemental appendices. We have added a sentence that prior antibiotic therapy was not an exclusion criteria.
Line 139: Authors have to specific the abbreviations ICSI and GUPI used in the table 2. They were not reported within the main text.
Reply: The clinical assessments are now spelled out in the main text (when first used) and in table 2.
Line 156: In the figure 2, authors have put in relevance the differences in mean relative abundance at the species levels between IC/BPS-Controls. Authors have described an higher abundance of Lactobacillus johnsonii as compared with the control group. However, several studies in rodents have shown the role of Lactobacillus johnsonii to prevent gastritis, reduce the level of pro-inflammatory cytokines, intestinal, extra-intestinal and systemic pro-inflammatory immune responses, diabetes. For this reason, I was wondering how the authors can interpret those results. Do they check the body weight, inflammatory markers, insulin and other parameters linked to metabolic and inflammatory disorders of those patients? Further information is needed.
Reply. We do not have the correlative data available for MAPP1. The ongoing 3-year MAPP2 study includes many of the variables mentioned by the review including inflammatory biomarkers and microbiome analysis of the stool and vaginal swabs. The analysis of this study will not be available for several years. This ongoing expanded study has been described in the discussion and conclusion sections
In addition, authors described a lower abundance of Corynebacterium in the IC/BPS participant population compared to control population. In the literature, other authors have described that Corynebacterium species are common in polymicrobial infections. Further info and explanation are needed.
Reply: We have revised that section as shown: “The finding of a decreased prevalence of Corynebacterium in the IC/BPS group may also be implicated. While it is difficult to explain any relevant mechanism one could hypothesize that Corynebacterium might be beneficial or even protective in the polymicrobial bladder microbiome of asymptomatic females.”
Line 158: To check the table 3, there is a mistake “9” has to be replaced with (.
Reply: This has been corrected in the revised manuscript.
Line 198: Authors have mentioned within the main text that a recent report observed microbiome changes in the stool of IC/BPS patients compared to asymptomatic controls. Why did not the authors decide to do the same analysis? Do they collect the stool material of those patients?
Reply: We did not collect stool in MAPP1. The NIH/NIDDK MAPP2 longitudinal study which is now ongoing does collect stool and vaginal specimens. Analysis will not be available for several years. This ongoing expanded study has been described in the discussion and conclusion sections.
The authors are grateful for the reviewers’ comments. The suggestions and recommendations have improved the quality of the manuscript.
Respectfully submitted:
J. Curtis Nickel MD (First Author)
Garth D. Ehrlich PhD (Senior Author)
Reviewer 2 Report
In this study, Curtis-Nickell et al., use ESI-TOF MS to identify microorganisms in patient urine samples. Comparing IC/BPS patients to healthy controls, the authors show no major differences in microbiota between the two groups other than some difference in Lactobacillus gasseri and Corynebacterium prevalence.
The introduction lacks sufficient background information on the urinary tract microbiota.
Figure 1.Shows 29 species unique to IC/BPS and 20 to controls. What are these species? This should be discussed.
Means species counts per group are reported as 2.5 and 2.3, how does this compare to other studies? Is this normal or is the technique used failing to identify species?
The presentation of figures is poor and needs to be improved
Author Response
Marta Varo, MSc
Genes Editorial Office, MDPI
Re: jcm-450936
Dear Ms. Varo,
The authors would like to thank the editors and the 4 reviewers for their careful and critical reviews of our manuscript jcm-450936 “A Culture-Independent Analysis of the Microbiota of Female Interstitial Cystitis/Bladder Pain Syndrome (IC/BPS) Participants in the MAPP Research Network”. We have answered each of the reviewer (and editors) queries below and have made the required changes in the revised manuscript (tracked in Word).
Reviewer 2:
In this study, Curtis-Nickell et al., use ESI-TOF MS to identify microorganisms in patient urine samples. Comparing IC/BPS patients to healthy controls, the authors show no major differences in microbiota between the two groups other than some difference in Lactobacillus gasseri and Corynebacterium prevalence.
The introduction lacks sufficient background information on the urinary tract microbiota.
Reply: We have expanded the introduction section to improve our description of both IC/BPS and the normal bladder microbiome (with additional appropriate references)
Figure 1.shows 29 species unique to IC/BPS and 20 to controls. What are these species? This should be discussed.
Reply: Yes, it would be a good idea to document the actual species noted in Figure 1. We have expanded the legend to Figure 1 to describe the species unique to IC/BPs, unique to controls and observed in both groups.
Means species counts per group are reported as 2.5 and 2.3, how does this compare to other studies? Is this normal or is the technique used failing to identify species?
Reply: The bladder microbiome has a considerably lower biomass than seen in bowel or even vagina and we believe this represents an accurate description of the bladder microbiome species.
The presentation of figures is poor and needs to be improved
Reply: We have improved the quality and resolution of the figures (2 and 3).
The authors are grateful for the reviewers’ comments. The suggestions and recommendations have improved the quality of the manuscript.
Respectfully submitted:
J. Curtis Nickel MD (First Author)
Garth D. Ehrlich PhD (Senior Author)
Reviewer 3 Report
Dear editor:
I appreciate the Editor to give me a chance to review this paper. This manuscript used PCR-ESI-TOF-MS to identify differences in the microbiota of the lower urinary tract between female IC/BPS and control participants. The result showed that Lactobacillus gasseri was slightly more prevalent in IC/BPS, while there was lower prevalence of Corynebacterium among IC/BPS participants compared with controls. Overall, the finding of the manuscript is interesting and the scientific premise is sound. However, some issues should be addressed.
1. The clinical significance of this finding in this work should be highlighted.
2. The titles of Sections 3,1, 3.2, 3.3 and 3.4 are missing.
3. In Line 146, the Table 2 should be Table 3.
4. In Table 3, 9SD) should be (SD).
5. The full names of abbreviation should be presented for the first time.
6. ‘compared to’ should be ‘compared with’.
7. The names of microbiota in species and genus levels should be italic in tables and figures.
Author Response
February 20, 2019
Marta Varo, MSc
Genes Editorial Office, MDPI
Re: jcm-450936
Dear Ms. Varo,
The authors would like to thank the editors and the 4 reviewers for their careful and critical reviews of our manuscript jcm-450936 “A Culture-Independent Analysis of the Microbiota of Female Interstitial Cystitis/Bladder Pain Syndrome (IC/BPS) Participants in the MAPP Research Network”. We have answered each of the reviewer (and editors) queries below and have made the required changes in the revised manuscript (tracked in Word).
Reviewer 3
I appreciate the Editor to give me a chance to review this paper. This manuscript used PCR-ESI-TOF-MS to identify differences in the microbiota of the lower urinary tract between female IC/BPS and control participants. The result showed that Lactobacillus gasseri was slightly more prevalent in IC/BPS, while there was lower prevalence of Corynebacterium among IC/BPS participants compared with controls. Overall, the finding of the manuscript is interesting and the scientific premise is sound. However, some issues should be addressed.
1. The clinical significance of this finding in this work should be highlighted.
Reply: We have expanded the discussion (second paragraph) and conclusion section to highlight the possible clinical potential of our findings.
2. The titles of Sections 3,1, 3.2, 3.3 and 3.4 are missing.
Reply: We have added appropriate titles to these sections
3. In Line 146, the Table 2 should be Table 3.
Reply: The reviewer is correct. We have revised that table reference to Table 3 in section 3.3. The editor also noted that Table 3 was not called out in the text. This is now corrected.
4. In Table 3, 9SD) should be (SD).
Reply: This has been corrected in the revised manuscript
5. The full names of abbreviation should be presented for the first time.
Reply: The full names have been added for each abbreviation the first time it is used in the manuscript
6. ‘compared to’ should be ‘compared with’.
Reply: We believe we have used “compared to” properly, but have revised according to recommendation of reviewer (we revised the abstract and discussion to “compared with”).
7. The names of microbiota in species and genus levels should be italic in tables and figures.
Reply: We have revised accordingly.
The authors are grateful for the reviewers’ comments. The suggestions and recommendations have improved the quality of the manuscript.
Respectfully submitted:
J. Curtis Nickel MD (First Author)
Garth D. Ehrlich PhD (Senior Author)
Reviewer 4 Report
The authors have performed a detailed analysis of microbiota from urinary tract of patients v control using culture independent methods. Although the results do not implicate any differences or causation/correlation with disease, yet this should be reported to help other researchers in the field to rule out microbiota dependent effects. The finding of Lactobacillus gasseri increase and Corynebacterium decrease can be important in future research
Author Response
February 20, 2019
Marta Varo, MSc
Genes Editorial Office, MDPI
Re: jcm-450936
Dear Ms. Varo,
The authors would like to thank the editors and the 4 reviewers for their careful and critical reviews of our manuscript jcm-450936 “A Culture-Independent Analysis of the Microbiota of Female Interstitial Cystitis/Bladder Pain Syndrome (IC/BPS) Participants in the MAPP Research Network”. We have answered each of the reviewer (and editors) queries below and have made the required changes in the revised manuscript (tracked in Word).
Reviewer #4:
Comments and Suggestions for Authors
The authors have performed a detailed analysis of microbiota from urinary tract of patients v control using culture independent methods. Although the results do not implicate any differences or causation/correlation with disease, yet this should be reported to help other researchers in the field to rule out microbiota dependent effects. The finding of Lactobacillus gasseri increase and Corynebacterium decrease can be important in future research
1. Does the introduction provide sufficient background and include all relevant references. – Can be improved.
Reply: We have now expanded the Introduction section (as per reviewer’s comments) to include a more detailed description of IC/BPS and the bladder microbiome including references.
The authors are grateful for the reviewers’ comments. The suggestions and recommendations have improved the quality of the manuscript.
Respectfully submitted:
J. Curtis Nickel MD (First Author)
Garth D. Ehrlich PhD (Senior Author)
Round 2
Reviewer 1 Report
There is a space more in the abstract, see the lines 34 and 36.
Line 63-64: It is not completely right, Enterobacteriaceae is a specific family of bacteria that can be associated with inflammatory disorders. In addition, this family has been associated in some animal rodents that develop tumors. So, I will pay more attention in saying bacteria are either non-pathogenic or even protective, it depends in the context. I suggest the authors to check this sentence.
Line 100-101: I really appreciate the huge amount of work that you have done but this is the major flaw of the entire study (previous antimicrobial therapy was not an exclusion criteria). The main goal of this study was to evaluate the gut microbiota of female interstitial Cystitis/Bladder Pain Syndrome (IC/BPS), and considering the important role of antibiotics in the depletion of bacteria, your analysis may not reflect the full picture of the microbiota of those patients. Most of the studies in humans have taken in consideration the intake of antibiotics.
Author Response
Reviewer #1
1. There is a space more in the abstract, see the lines 34 and 36.
Reply: The two spaces have been removed.
2. Line 63-64: It is not completely right, Enterobacteriaceae is a specific family of bacteria that can be associated with inflammatory disorders. In addition, this family has been associated in some animal rodents that develop tumors. So, I will pay more attention in saying bacteria are either non-pathogenic or even protective, it depends in the context. I suggest the authors to check this sentence.
Reply: The point we were trying to make is that Enterobacteriaceae which is regarded as uropathogenic, can be identified in healthy bladders of asymptomatic individuals. We have rewritten that section to make sure the intent of that observation is clear and decided to delete the discussion of possibly protective influence. There is little or no clinical evidence that Enterobacteriaceae cause bladder tumors (unless patient develops squamous metaplasia secondary to long term overt chronic infection). A complicated discussion on the role of bacteria in urinary tract carcinogenesis is beyond the scope of this paper and not really relevant to this study.
3. Line 100-101: I really appreciate the huge amount of work that you have done but this is the major flaw of the entire study (previous antimicrobial therapy was not an exclusion criteria). The main goal of this study was to evaluate the gut microbiota of female interstitial Cystitis/Bladder Pain Syndrome (IC/BPS), and considering the important role of antibiotics in the depletion of bacteria, your analysis may not reflect the full picture of the microbiota of those patients. Most of the studies in humans have taken in consideration the intake of antibiotics.
Reply: As stated in the the first line of the abstract and the last line of the Introduction section, the main goal of this study, contrary to the reviewer’s impression, was to contrast the differences between the lower urinary tract microbiota between subjects who have a diagnosis of IC/BPS and a control population of females with no diagnosis or symptoms of IC/BPS and not to evaluate the gut microbiota of female IC/BPS subjects. That being said, the lack of data regarding antibiotic use is clearly a limitation and we have added a sentence in the Discussion in this regard. The goal stated by the reviewer is in fact one of the main goals of our ongoing MAPP 2 study described in the last sentence of the Discussion section (to include the gut and vaginal microbiota) and in that study antibiotic exposure will be determined a priori. We have noted this in this revision.
Reviewer 2 Report
Thank you for submitting a revised manuscript. The species unique to IC/BPS revealed some interesting bacteria.
Francisella tularensis is potentially a highly pathogenic bacterium which can be lethal.
Helicobacter is another species with clear pathogenic potential.
Candida species are also of interest. Microbiota dysbiosis is often thought to predispose to fungal disease.
Proteus mirabilis is another species with links to pathology.
Is anything known about interactions between Corynebacterium and these interesting species?
A little more discussion on these species would be appreciated.
Author Response
Reviewer #2
Thank you for submitting a revised manuscript. The species unique to IC/BPS revealed some interesting bacteria.
1. Francisella tularensis is potentially a highly pathogenic bacterium which can be lethal.Helicobacter is another species with clear pathogenic potential.Candida species are also of interest. Microbiota dysbiosis is often thought to predispose to fungal disease. Proteus mirabilis is another species with links to pathology.
Reply: As well as the organisms described by the reviewer, there are a number of organisms unique to the IC/BPS group that could be considered potentially uropathogenic. As suggested in the reviewer’s 3rd comment we have added more discussion about these potentially pathogenic organisms found to be unique in the IC/BPS group (and revised the conclusions accordingly).
2. Is anything known about interactions between Corynebacterium and these interesting species?
Reply: The authors did add a discussion of the possible role of Corynebacterium in the revision. Because Cornyebacterium sp. were not unique to our IC/BPS patient population and we actually found a lower prevalence in the IC/BPS group compared to the control group, we did not believe that we could expand this discussion.
3. A little more discussion on these species would be appreciated.
Reply: As discussed in the reply to the first comment of this reviewer, we have added further to the discussion section (and noted in the conclusions) in regard to the unique, potentially pathogenic bacteria observed in the IC/BPS group.